# A New Approach for Increasing Speed, Loading Capacity, Resolution, and Scalability of Preparative Size-Exclusion Chromatography of Proteins

**Yating Xu**, **Si Pan** and **Raja Ghosh** *

Department of Chemical Engineering, McMaster University, Hamilton, ON L8S 4L7, Canada
* Correspondence: rghosh@mcmaster.ca; Tel.: +1-905-525-9140 (ext. 27415)

**Abstract:** Low speed, low capacity, and poor scalability make size-exclusion chromatography (SEC) unattractive for use in the preparative separation of proteins. We discuss a novel $z^2$ cuboid SEC device that addresses these challenges. A $z^2$ cuboid SEC device (~24 mL volume) was systematically compared with a conventional SEC column having the same volume and packed with the same resin. The primary objective of this study was to use the same volume of SEC medium in a much more efficient way by using the novel device. At any given flow rate, the pressure drop across the $z^2$ cuboid SEC device was lower by a factor of 6 to 8 due to its shorter bed height and greater cross-sectional area. Under overloaded conditions, the peaks obtained during protein separation with the conventional column were poorly resolved and showed significant fronting, while those obtained with the $z^2$ cuboid SEC device were much better resolved and showed no fronting. At any given flow rate, better resolution was obtained with the $z^2$ cuboid SEC device, while for obtaining a comparable resolution, the flow rate that could be used with the $z^2$ cuboid SEC device was higher by a factor of 2 to 3. Hence, productivity in SEC could easily be increased by 200 to 300% using the $z^2$ cuboid SEC device. The scalability of the $z^2$ cuboid SEC device was also demonstrated based on a device with a 200 mL bed volume.

**Keywords:** chromatography; protein separation; size-exclusion chromatography; cuboid packed bed; loading capacity; scalability

## 1. Introduction

Size-exclusion chromatography (SEC) is widely used for the analysis of proteins [1–3] and many other types of macromolecules such as DNA [4], RNA [5], and carbohydrates [6]. The main reason why SEC is so widely used in the fields of analytical chemistry and biochemistry is that it utilizes one of the most fundamental physicochemical properties of a molecule, i.e., its molecular weight (or mass) for separation [1,2]. Most other chromatographic separation techniques, such as ion-exchange chromatography (IEC) [7], hydrophobic interaction chromatography (HIC) [8], and reverse-phase chromatography (RPC) [9], utilize dynamic properties of molecules such as surface charge and surface hydrophobicity, which depend on solution conditions such as pH and ionic strength. While the use of such dynamic properties is advantageous from the point of view of process flexibility and optimization, the interpretation of their underlying mechanisms can be quite challenging [7,10–13]. Consequently, SEC is the primary method for the chromatographic analysis of proteins and other biomacromolecules [14,15], and methods such as IEC, HIC, and RPC are mainly used as secondary analytical tools. Despite such extensive applications in analytical separations, SEC has relatively limited use in the preparative (or process) chromatography of proteins [16,17]. The three main limitations of SEC that restrict its usage in preparative chromatography are low speed, low capacity, and poor scalability [16,17]. Consequently, the main established application of SEC in preparative biopharmaceutical purification is

desalting, i.e., the removal of salts and other low-molecular-weight species from protein solutions and solutions of biomacromolecules in general [16–19].

The resolution of separated peaks in SEC is directly linked to the speed of separation, i.e., good resolutions can only be obtained at very painfully low flow rates. This can be explained by invoking the van Deemter equation [20]:

$$h = A + B/u + Cu \tag{1}$$

where $h$ is the plate height, $A$ is the eddy dispersion term, $B/u$ is the axial diffusion term, $Cu$ is the resistance to transfer term, and $u$ is the superficial (or linear) velocity. The resolution depends on $h$, i.e., better resolution is obtained at lower values of $h$. As evident from Equation (1), the value of $h$ depends on the system constants $A$, $B$, and $C$, and on the value of $u$. The value of $A$ depends on the nature of the stationary phase particles and is not significantly influenced by operating conditions. For macromolecules such as proteins, the contribution of the second term in Equation (1), i.e., $B/u$, is relatively insignificant, except at extremely low flow rates. In protein chromatography, the resolution is very significantly influenced by the third term in Equation (1), i.e., $Cu$, as the transport of macromolecules within the pores present in the stationary phase particles is typically very slow. Increasing the value of $u$ increases the solute band width within the column due the greater imbalance between intrapore and axial solute transport [21]. In other words, as the flow rate increases, the value of $Cu$ increases, the value of $h$ increases, and the resolution decreases. Therefore, SEC needs to be carried out at very low flow rates to obtain satisfactory resolutions [16]. Increasing the flow rate in SEC can also result in problems such as peak fronting and peak tailing [2,22,23], both of which could result in peak broadening and ultimately poor resolution, increase in eluate pool volume, and the dilution of the eluate. Therefore, both recovery and purity can be adversely affected due to increases in flow rate. On the other hand, operating the separation process at a low flow rate decreases productivity and increases the likelihood of product degradation during the separation process [24].

The current approach for designing size-exclusion chromatography columns is to use very high length-to-diameter ($L/d$) ratios [2,25–29]. Although this results in high resolutions in separation [2,25,27], long columns generate considerable back pressure, even at quite low flow rates [28]. This restricts the speed of separation and imposes severe limits on scaling-up. High pressure drop can also result in other problems such as resin compaction [30,31]. The problem of high pressure drop could be addressed by using a shorter but wider column having the same volume ($V$), i.e., one with a lower length ($L$)-to-diameter ($d$) ratio. If both columns are operated at the same flow rate ($Q$), the shorter column, due to the combination of its shorter length and larger area of cross section ($A$), would have a significantly lower pressure drop ($P$). However, as extensively discussed in the literature [2,25–29], short SEC columns give extremely poor resolution ($R$) when compared to long conventional SEC columns. The poor separation performance obtained with a short and wide SEC column would make any advantage gained in terms of lower back pressure quite meaningless. Therefore, not surprisingly, short and wide columns are not used for carrying out SEC.

In a recent paper [32], the use of a flat cuboid packed-bed chromatography device with a short bed height for the high-resolution ion-exchange separation of proteins was discussed. This device was provided with complimentary lateral channels for efficient flow distribution and collection from a box-shaped resin-packed bed. A head-to-head comparison with a tall column with the same bed volume and packed with the same resin medium showed that the flat cuboid packed-bed chromatography device outperformed the tall column in terms of every aspect that was compared, i.e., it could be operated at a far lower back pressure and gave a superior resolution [32]. More recently, the use of a $z^2$ cuboid packed-bed device, which is an improved version of the older cuboid packed-bed device, for hydroxyapatite nanoparticle-based protein separation was discussed [33]. The design features of the cuboid packed-bed device [32] and the $z^2$ cuboid packed-bed device [33] minimized the extent of

macroscale convective dispersion [34] within these devices, resulting in superior separation performances compared to their equivalent conventional columns. This approach for minimizing macroscale convective dispersion aligns well with one of the basic principles of process intensification [35], which states that the efficiency of a flow-based chemical reaction or separation process can be maximized if the molecules being separated are treated more equally, i.e., if their experiences during their transit through the reaction or separation device are similar. Moreover, due to their wider areas of cross section (compared to their equivalent control columns), these devices could be operated at very low superficial velocities. By doing so, the third term on the right-hand side of Equation (1), i.e., $Cu$, could be minimized, and ultimately the value of $h$ could also be minimized. This was one of the principal reasons why the resolution obtained with low-bed-height cuboid chromatography devices was very high [32–34].

The primary objective of this study was to use the same volume of SEC medium in a much more efficient way by using the $z^2$ cuboid SEC device. To demonstrate the role of bed height and column diameter in cylindrical columns, a short and wide SEC column was first compared with a conventional SEC column (both with 24 mL volumes and packed with Sephacryl S-200 HR medium) based on the resolution obtained in binary protein separation. The potential for using a $z^2$ cuboid packed-bed device with a short bed height for the size-exclusion chromatography of proteins was then examined. The performance of a $z^2$ cuboid packed-bed device with a bed volume of 24 mL and packed with Sephacryl S-200 HR medium was compared with that of a 24 mL commercial SEC column packed with the same medium. These devices were first compared in terms of their pressure drop at different flow rates. The speed of separation and the resolution obtained in binary model protein fractionation were then systematically compared. The scalability of the $z^2$ cuboid SEC device was demonstrated through experiments carried out using a larger (200 mL bed volume) device. The obtained results are discussed.

## 2. Materials and Methods

### 2.1. Materials

Bovine serum albumin (A9418) and lysozyme (L6876) were purchased from Sigma-Aldrich (St. Louis, MO, USA). Phosphate-buffered saline (pH 7.4), or PBS, was made using sodium phosphate dibasic (S0876), potassium chloride (P9541), potassium phosphate monobasic (P5655), and sodium chloride (S7653) purchased from Sigma-Aldrich (St. Louis, MO, USA). A Tricorn GL 10/300 chromatography column (10 mm diameter, 300 mm height, label # 71-5019-07-EG) and Sephacryl S-200 HR SEC medium (GE17-0584-10) were purchased from GE Healthcare Bioscience (Uppsala, Sweden). Hydrophilic MCE membrane (SCWP04700, 8.0 μm pore size) was purchased from Millipore (Billerica, MA, USA). All buffers and sample solutions were prepared using ultrapure water (18.2 MΩ cm) obtained from a SIMPLICITY 185 water purification unit (Millipore, Molsheim, France). All buffers were degassed and microfiltered using a PVDF membrane (VVLP04700, 0.1 μm pore size, Millipore, Billerica, MA, USA) just prior to the chromatography experiments.

### 2.2. Short and Wide Column

The conventional Tricorn GL 10/300 column, which was used as the control chromatography device, had a diameter of 10 mm (area of cross section = 0.785 cm$^2$), and if packed to a height of 300 mm, had an effective volume of about 24 mL. The short and wide column used in this study had the same bed volume (i.e., 24 mL), a bed height of 60 mm, and a diameter 22.6 mm (area of cross section = 4 cm$^2$). Therefore, it was shorter than the conventional Tricorn GL 10/300 by a factor of 5. The short and wide column consisted of the columnar section of the device, which housed the resin bed, flanked by the two headers, each of which were provided with radial channels similar to those in the Tricorn GL 10/300 column for flow distribution. All parts of the short and wide column were printed with Photopolymer Resin Clear (FLGPCL04, Formlabs, Somerville, MA, USA) using a Form 2 3D printer (Formlabs, Somerville, MA, USA). This column was packed

with Sephacryl S-200 HR resin by slurry packing with a protocol similar to that used to pack the Tricorn GL 10/300 column. The resin was retained within the column using two discs of hydrophilic MCE membrane (8.0 μm pore size), one below and one above the columnar section. Each membrane was sandwiched between two thin layers of gaskets to prevent leakage.

### 2.3. $z^2$ cuboid SEC Devices

Figure 1 shows the layout of primary and secondary channels in a $z^2$ cuboid packed-bed device [33,34]. The feed is introduced into the device through a primary channel on the top, located at a slight offset from the top plane of the packed bed, from where it is distributed to the cuboid packed bed through a set of secondary distribution channels located above the packed bed. The liquid is collected from the packed bed using a set of secondary collection channels located at the bottom of the device, and from there the liquid is drained through a primary collection channel to the outlet. The $z^2$ cuboid SEC devices used in this study consisted of three main parts (see Figure 2): a top distribution plate, a cuboid frame for housing the cuboid packed bed, and a bottom collection plate. The design of the collection plates was based on the mirror reflection of the distribution plate [33,34]. These parts were 3D-printed with a Form 2 3D printer using Photopolymer Resin Clear. The top and bottom plates were provided with primary and secondary channels (see the bottom plate in Figure 2), and the arrangement of channels in the two plates was complimentary to ensure the characteristic $z^2$ flow distribution and collection [33,34]. Two $z^2$ cuboid SEC devices were fabricated for this project, one small (~24 mL volume) and one big (~200 mL volume). The 24 mL $z^2$ cuboid device had a five times larger cross-sectional area (4 cm$^2$, 20 mm × 20 mm) and a five times smaller bed height of 60 mm compared to the conventional SEC column (Tricorn GL 10/300), which was used as a control device. The primary channels in the 24 mL $z^2$ cuboid device had a circular cross section of 1 mm diameter and were embedded within the plates, while the secondary channels (15 in number) were segment-shaped with 1 mm width and 0.25 mm depth, i.e., open on the flat side. Rubber gaskets were placed between the frame and the plates on both sides to prevent leakage. On each side, a square piece of hydrophilic MCE membrane (8.0 μm pore size) was placed between the gasket and the plate to retain the resin particles within the device. These membranes also helped to create segment-shaped secondary channels for flow distribution and collection. An image of the assembled 24 mL $z^2$ cuboid SEC device is shown in Figure 3A. As indicated in the figure, the inlet and outlet were located at space diagonally opposite ends. The 200 mL $z^2$ cuboid device had an area of cross section of 16 cm$^2$ (40 mm × 40 mm) and a bed height of 125 mm. Therefore, the length, width, and height of the 200 mL $z^2$ cuboid SEC device were each approximately double the corresponding dimensions of the 24 mL $z^2$ cuboid SEC device. Therefore, the one-dimensional scale-up factor was about 2, while the volumetric scale-up factor was about 8. The two plates in the 200 mL $z^2$ cuboid SEC device contained embedded primary and secondary channels (3 mm and 2 mm diameters, respectively). The resin was retained within the frame using a nylon mesh. An image of the assembled 200 mL $z^2$ cuboid SEC device is shown in Figure 3B. Sephacryl S-200 HR resin was packed within the two $z^2$ cuboid SEC devices by slurry packing.

### 2.4. Size-Exclusion Chromatography Experiments

Size-exclusion chromatography experiments were carried out with mixtures of model proteins (BSA and lysozyme) as feed solution. The experiments with the 25 mL devices (i.e., the conventional and short and wide columns, and the 24 mL $z^2$ cuboid SEC device) were carried out using an AKTA Prime Plus liquid chromatography system (GE Healthcare Biosciences, Montreal, QC, Canada). The experiments with the 200 mL $z^2$ cuboid SEC device were carried out using an NGC Quest 100 Chromatography System (Bio-Rad Laboratories Canada Ltd., Mississauga, ON, Canada). Phosphate-buffered saline (pH 7.4), or PBS, was used as the mobile phase and for preparing feed solutions. Before carrying out experiments

with the $z^2$ cuboid SEC devices, 20% ethanol in water was used to prime the channels, i.e., to remove air bubbles.

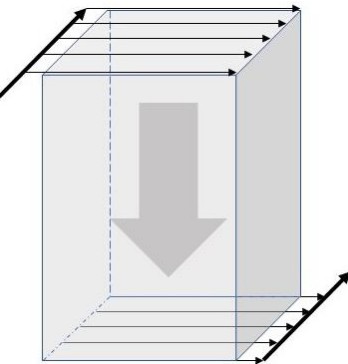

**Figure 1.** Schematic diagram showing flow channels (indicated by arrows) in a $z^2$ cuboid packed-bed device.

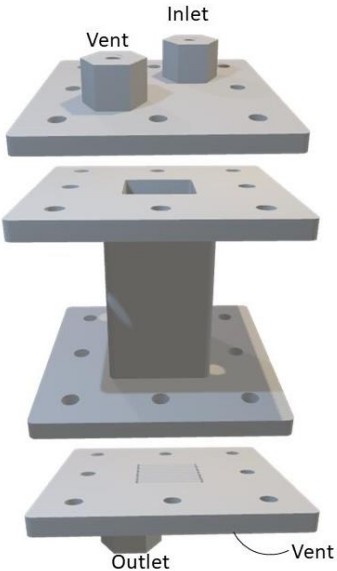

**Figure 2.** Different components of a $z^2$ cuboid SEC device.

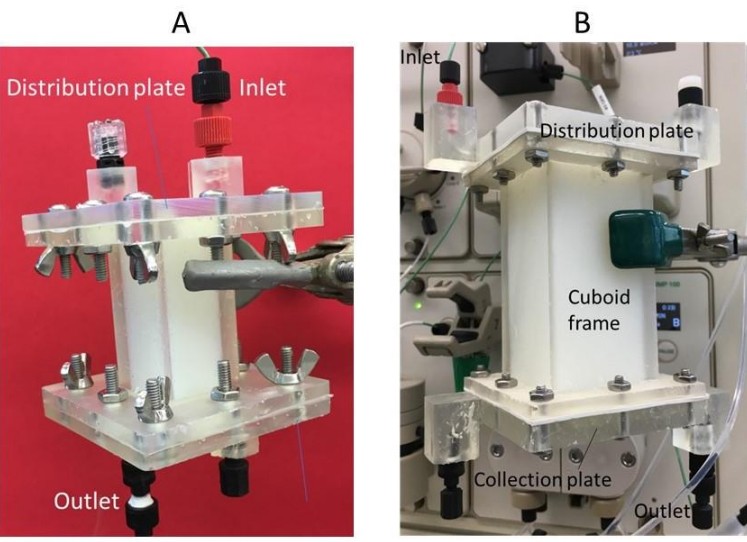

**Figure 3.** (**A**) Photograph of the 24 mL $z^2$ cuboid SEC device used in this study. (**B**) Photograph of the 200 mL $z^2$ cuboid SEC device used in this study.

## 3. Results and Discussion

### 3.1. Comparison of Two Columns

First, the performance of the conventional (Tricorn GL 10/300) SEC column was compared with that of the short and wide column by carrying out the separation of a BSA–lysozyme mixture. In these experiments, which were carried out at a flow rate of 0.5 mL/min (the manufacturer's recommended flow rate for the Tricorn GL 10/300 column), 2 mL of feed sample was injected. The feed volume being close to 10% of the column volume, these separation experiments represented a "high-loading" condition. Figure 4A shows the chromatogram obtained with the Tricorn GL 10/300 column, while Figure 4B shows the corresponding chromatogram obtained with the short and wide column. As expected, the separation of BSA and lysozyme obtained with the conventional (Tricorn GL 10/300) column was far superior to that obtained with the short and wide column. To begin with, the protein peaks obtained with the latter were wider. This was particularly the case with the second (i.e., lysozyme) peak, which showed two frontings, the first around 22 to 23 mL and the second around 27 to 28 mL of effluent volume. Some fronting could also be observed in the lysozyme peak obtained with the conventional (Tricorn GL 10/300) column. The broadening of the lysozyme peak in the short and wide column also explained why its height was significantly lower than the peak obtained with the Tricorn GL 10/300 column. The resolution (*R*) of well-resolved peaks is typically measured based on the peak width at 4.4% of the peak height (the five-sigma method) [36]. However, in the chromatogram obtained with the short and wide column, the valley between the BSA and lysozyme peaks was well above 4.4% of the height of the lysozyme peak. The BSA and lysozyme peak widths at half height were 3.14 and 5.85 mL, and 3.23 and 7.87 mL, respectively, for the Tricorn GL 10/300 and short and wide columns.

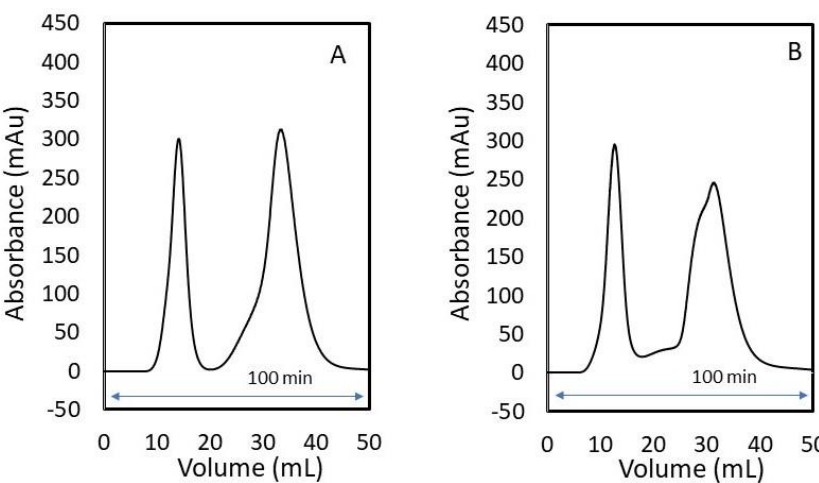

**Figure 4.** Chromatograms obtained during the separation of BSA (5 g/L) and lysozyme (2.5 g/L) using the (**A**) conventional (Tricorn GL 10/300) column and (**B**) the short and wide column (medium: Sephacryl S-200 HR; mobile phase: phosphate-buffered saline (pH 7.4); flow rate: 0.5 mL/min; loop size: 2 mL).

These results clearly demonstrate that a short and wide column gives significantly poorer separation in size-exclusion chromatography than a conventional tall SEC column having the same volume. As discussed in several papers [32–34], the performance of short and wide columns is adversely affected by macroscale convective dispersion. In earlier studies such effect was demonstrated in ion-exchange [34] and mixed-mode separation [33]. The results shown in Figure 4 clearly demonstrate the significant impact of macroscale convective dispersion in wide SEC columns. These results also explain why short and wide columns are not used to carry out size-exclusion chromatography.

### 3.2. Comparison of 24 mL Conventional Column with 24 mL z² cuboid SEC Device

After demonstrating that a short and wide column would not be suitable for carrying out size-exclusion chromatography, experiments were carried out to examine whether a short and wide $z^2$ cuboid SEC device would be an option. Earlier studies have shown that the extent of macroscale convective dispersion could be suppressed by the flow distribution and collection features of this type of device [33,34].

A systematic comparison was carried out, using the conventional (Tricorn GL 10/300) column as the control device. Table 1 shows the pressure drop data obtained with the Sephacryl S-200 HR medium packed Tricorn GL 10/300 column and the 24 mL $z^2$ cuboid SEC device using PBS as the mobile phase. At a given flow rate, the pressure drop across the 24 mL $z^2$ cuboid SEC device was lower than that across the Tricorn GL 10/300 column by a factor of 6 to 8. Therefore, even though both devices had the same bed volume, the $z^2$ cuboid SEC device, due to its different geometry, could be operated at a significantly higher flow rate than that possible with the column. Notably, with the conventional (Tricorn GL 10/300) column, significant medium packing (as evident from the creation of gaps at the top of the column) and the resultant decrease in permeability were observed at flow rates higher than 2 mL/min. Moreover, at these higher flow rates, the pressure drop increased with time. With the 24 mL $z^2$ cuboid SEC device, experiments could be carried out at flow rates as high as 5 mL/min without any of the above-mentioned adverse effects.

**Table 1.** Pressure drop at different flow rates for Sephacryl S-200 HR medium packed 24 mL Tricorn GL 10/300 column and 24 mL $z^2$ cuboid SEC device (mobile phase: phosphate-buffered saline, pH 7.4).

| Flow Rate (mL/min) | Column Pressure (MPa) | $z^2$ cuboid Pressure (MPa) |
|:---:|:---:|:---:|
| 0.5 | 0.035 | 0.006 |
| 1.0 | 0.072 | 0.010 |
| 1.5 | 0.114 | 0.016 |
| 2.0 | 0.168 | 0.021 |
| 2.5 | - | 0.026 |
| 3.0 | - | 0.032 |
| 5.0 | - | 0.051 |

Figure 5 shows the chromatograms obtained during BSA (5 mg/mL)–lysozyme (2.5 mg/mL) separation using the Sephacryl S-200 HR medium packed Tricorn GL 10/300 column (Figure 5A) and the 24 mL $z^2$ cuboid SEC device (Figure 5B) at a flow rate of 0.5 mL/min. The volume of sample injected in these experiments was 2 mL (~8.3% of bed volume). The BSA and lysozyme peaks were resolved very close to the baseline using both devices. However, the lysozyme peak obtained with the conventional (Tricorn GL 10/300) column had significant fronting due to the use of sample overload condition. The BSA and lysozyme peaks obtained with the 24 mL $z^2$ cuboid SEC device were both sharper and narrower than those obtained using the Tricorn GL 10/300 column and did not show any fronting or tailing. The difference in the peak retention volume was slightly greater in the chromatogram obtained using the Tricorn GL 10/300 column. The BSA (5 mg/mL)–lysozyme (2.5 mg/mL) mixture was also separated using the 24 mL $z^2$ cuboid SEC device at higher flow rates, i.e., 1.0, 1.5, 2.0, 2.5, 3.0, and 5.0 mL/min (see Figure 6). As the flow rate increased, the separation time and the resolution decreased (as evident by visual inspection). Since the peaks were resolved close to the baseline, the resolution factor (*R*) was calculated using the five-sigma method [36] using the equation shown below:

$$R = \frac{2.5(V_A - V_B)}{w_{4.4\ A} + w_{4.4\ B}} \quad (2)$$

where $w_{4.4}$ is the peak width at 4.4% of the maximum peak height, $V$ is the peak retention volume, and subscripts *A* and *B* refer to the second and the first eluted peaks, respectively. The resolution factor data obtained from the above BSA–lysozyme separation experiments

are summarized in Table 2. At a flow rate of 0.5 mL/min, the resolution obtained using the 24 mL $z^2$ cuboid SEC device was 30% higher than that obtained using the conventional (Tricorn GL 10/300) column. At the same flow rate, the superficial velocity within the 24 mL $z^2$ cuboid SEC device was lower than that in the column by a factor of five. Hence, the extent of dispersion due to the resistance to mass transfer (i.e., the *Cu* term in Equation (1)) could be expected to be significantly lower in the former. This effect, albeit in bind-and-elute separations using ion-exchange medium, has been discussed previously [32]. Moreover, the flow distribution and collection features of a $z^2$ cuboid SEC device minimized the extent of macroscale convective dispersion [33,34]. The resolution obtained with the 24 mL $z^2$ cuboid SEC device at a flow rate of 1.5 mL/min was comparable with that obtained using the Tricorn GL 10/300 column at 0.5 mL/min. Based on this comparison, it may be reasoned that to obtain comparable separation (in terms of resolution factor) the process could be sped up by 300% using the $z^2$ cuboid SEC device.

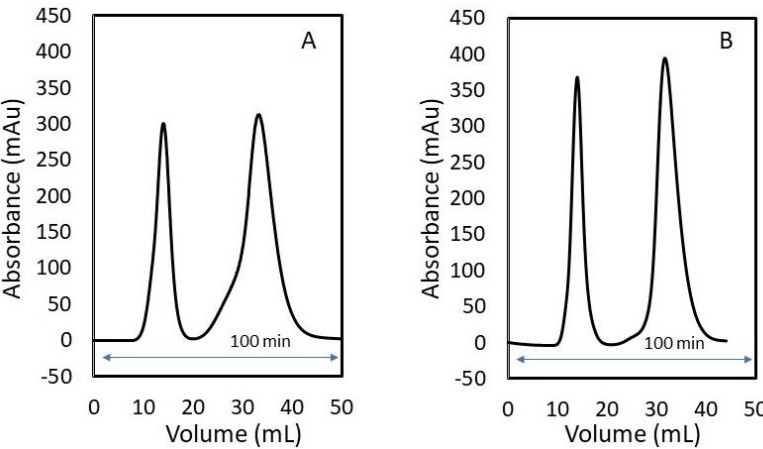

**Figure 5.** Chromatograms obtained during BSA (5 mg/mL) and lysozyme (2.5 mg/mL) separation using (**A**) the conventional (Tricorn GL 10/300) column and (**B**) the 24 mL $z^2$ cuboid SEC device (medium: Sephacryl S-200 HR; mobile phase: phosphate-buffered saline (pH 7.4); flow rate: 0.5 mL/min; loop size: 2 mL).

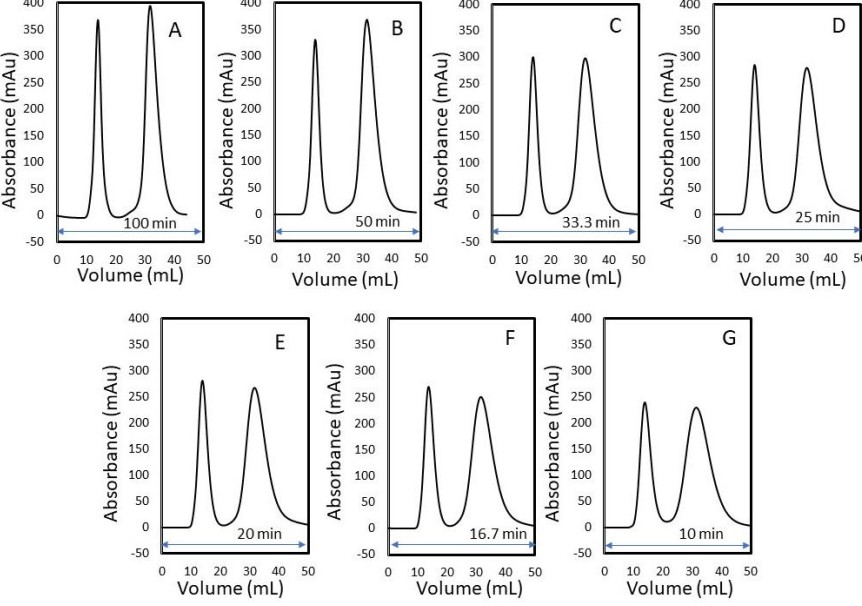

**Figure 6.** Comparison of chromatograms obtained during BSA (5 mg/mL) and lysozyme (2.5 mg/mL) separation using the 24 mL $z^2$ cuboid SEC device at different flow rates (medium: Sephacryl S-200 HR; mobile phase: phosphate-buffered saline (pH 7.4); loop size: 2 mL): (**A**) 0.5 mL/min, (**B**) 1.0 mL/min, (**C**) 1.5 mL/min, (**D**) 2 mL/min; (**E**) 2.5 mL/min; (**F**) 3 mL/min; (**G**) 5 mL/min.

**Table 2.** Resolution in BSA–lysozyme separation using Sephacryl S-200 HR medium packed 24 mL Tricorn GL 10/300 column and 24 mL $z^2$ cuboid SEC device (mobile phase: phosphate-buffered saline, pH 7.4; loop: 2 mL; feed: 5 mg/mL BSA + 2.5 mg/mL lysozyme).

| Device | Flow Rate (mL/min) | BSA Peak Width (mL) | Lysozyme Peak Width (mL) | BSA Peak Retention Volume (mL) | Lysozyme Peak Retention Volume (mL) | Resolution |
|---|---|---|---|---|---|---|
| Column | 0.5 | 8.072 | 19.990 | 13.923 | 34.076 | 1.80 |
| $z^2$ cuboid | 0.5 | 6.551 | 12.351 | 13.951 | 31.703 | 2.35 |
| $z^2$ cuboid | 1.0 | 7.256 | 15.271 | 13.917 | 31.569 | 1.96 |
| $z^2$ cuboid | 1.5 | 7.874 | 17.491 | 13.891 | 31.720 | 1.76 |
| $z^2$ cuboid | 2.0 | 8.341 | 21.016 | 13.758 | 31.687 | 1.53 |
| $z^2$ cuboid | 2.5 | 8.593 | 21.310 | 13.806 | 31.736 | 1.50 |
| $z^2$ cuboid | 3.0 | 8.718 | 21.709 | 13.835 | 31.664 | 1.46 |
| $z^2$ cuboid | 5.0 | N/A * | N/A * | 13.704 | 31.271 | N/A * |

* N/A: Peaks resolved above 4.4% of peak height. Therefore, peak width at this height could not be measured.

Separation experiments with even higher BSA–lysozyme loadings were carried out by increasing the loop volume to 5 mL (~20.8% of bed volume). The typical recommended column loading in preparative size-exclusion chromatography is 5 to 10% of the column volume [2]. The sample loading in the current experiment was therefore well in excess of the typical recommended limit. Figure 7A shows the chromatograms obtained with the Tricorn GL 10/300 column at 0.5 mL/min, while Figure 7B shows the chromatograms obtained with the 24 mL $z^2$ cuboid SEC device at twice that flow rate, i.e., 1.0 mL/min. As observed with the chromatogram obtained by 2 mL sample injection (see Figure 5), the lysozyme peak in the chromatogram obtained using the Tricorn GL 10/300 column with 5 mL sample injection had very significant fronting. No fronting was observed in the lysozyme peak obtained with the 24 mL $z^2$ cuboid SEC device. Table 3 summarizes the resolution factors obtained from the experiments carried out using 5 mL sample injection. These results show that a comparable resolution in BSA–lysozyme separation could be obtained using the 24 mL $z^2$ cuboid SEC device at double the flow rate used with the 24 mL Tricorn GL 10/300 column, i.e., the separation could be sped up by 200%.

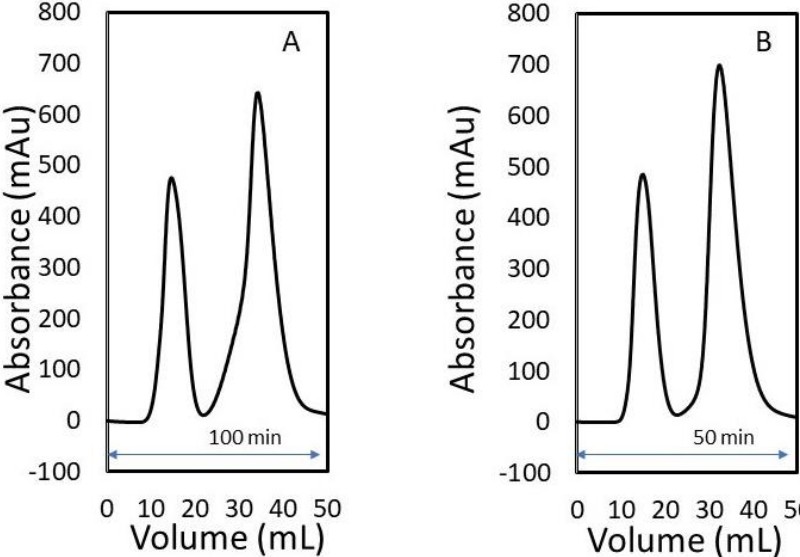

**Figure 7.** Chromatograms obtained during BSA (5 mg/mL) and lysozyme (2.5 mg/mL) separation using (**A**) the 24 mL Tricorn GL 10/300 column at a 0.5 mL/min flow rate and (**B**) the 24 mL $z^2$ cuboid SEC device at a 1 mL/min flow rate (medium: Sephacryl S-200 HR; mobile phase: phosphate-buffered saline (pH 7.4); loop size: 5 mL).

**Table 3.** Resolution in BSA–lysozyme separation using Sephacryl S-200 HR medium packed 24 mL Tricorn GL 10/300 column and 24 mL $z^2$ cuboid SEC device (mobile phase: phosphate-buffered saline, pH 7.4; loop: 5 mL; feed: 5 mg/mL BSA + 2.5 mg/mL lysozyme).

| Device | Flow Rate (mL/min) | BSA Peak Width (mL) | Lysozyme Peak Width (mL) | BSA Peak Retention Volume (mL) | Lysozyme Peak Retention Volume (mL) | Resolution |
|---|---|---|---|---|---|---|
| Column | 0.5 | 10.781 | 20.965 | 14.518 | 34.291 | 1.56 |
| $z^2$ cuboid | 1.0 | 10.831 | 17.979 | 15.055 | 32.329 | 1.50 |

In order to further challenge the loading capacity of the 24 mL $z^2$ cuboid SEC device, the amount of sample injected was increased from 5 mL to 7.5 mL (~31.3% of the bed volume). Figure 8 shows the chromatograms obtained during separation at 0.5 and 1.0 mL/min, and the data obtained from these experiments are summarized in Table 4. As expected, the resolution decreased a bit with the increase in sample loading. However, even at a 1.5 mL/min flow rate with 7.5 mL sample loading, the BSA and lysozyme peaks were resolved close to the baseline.

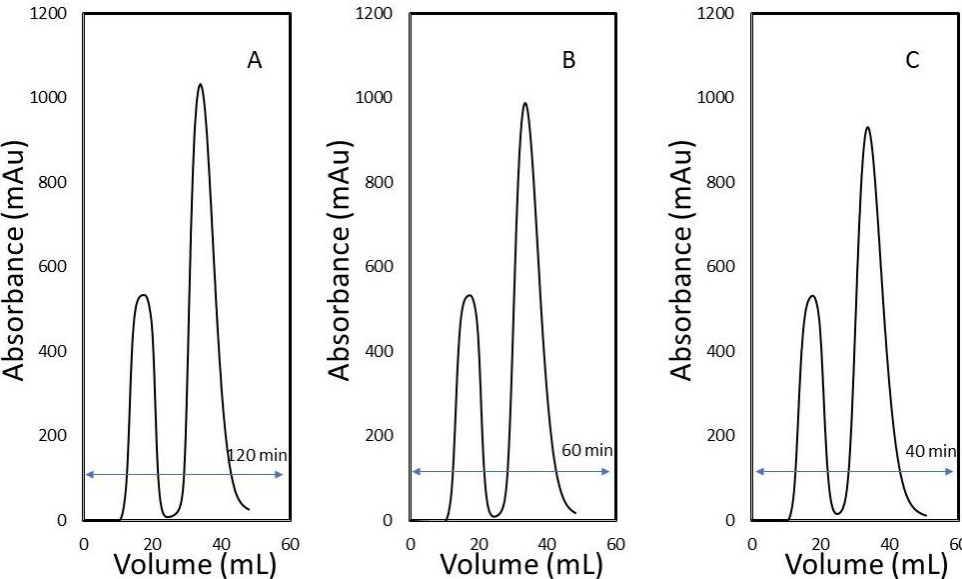

**Figure 8.** Chromatograms obtained during BSA (5 mg/mL) and lysozyme (2.5 mg/mL) separation using 24 mL $z^2$ cuboid SEC device by loading 7.5 mL of feed sample at different flow rates: (**A**) 0.5 mL/min, (**B**) 1 mL/min, (**C**) 1.5 mL/min (medium: Sephacryl S-200 HR; mobile phase: phosphate-buffered saline (pH 7.4)).

**Table 4.** Resolution in BSA–lysozyme separation using Sephacryl S-200 HR medium in 24 mL $z^2$ cuboid packed-bed device (mobile phase: phosphate-buffered saline, pH 7.4; loop: 7.5 mL; feed: 5 mg/mL BSA + 2.5 mg/mL lysozyme).

| Flow Rate (mL/min) | BSA Peak Width (mL) | Lysozyme Peak Width (mL) | BSA Peak Retention Volume (mL) | Lysozyme Peak Retention Volume (mL) | Resolution |
|---|---|---|---|---|---|
| 0.5 | 12.206 | 17.010 | 17.605 | 33.857 | 1.46 |
| 1.0 | 11.644 | 17.736 | 17.438 | 33.685 | 1.38 |
| 1.5 | 12.206 | 19.011 | 17.605 | 33.635 | 1.28 |

### 3.3. Scale-Up Experiments with 200 mL $z^2$ cuboid SEC Device

Protein purification experiments were carried out using the 200 mL $z^2$ cuboid SEC device to assess scalability. In the initial experiments, the feed composition was 5 g/L BSA, and 2.5 g/L lysozyme prepared in PBS. Figure 9 shows a representative chromatogram obtained at a flow rate of 8 mL/min by injecting 40 mL of feed sample (~20% of the bed volume). These experiments were carried out on three consecutive days (nine experiments in total) to check for intraday and interday variability. In all these experiments, the BSA and lysozyme peaks were resolved at the baseline ($R = 1.58 \pm 0.06$). The average asymmetry factors for the BSA and lysozyme peaks were 1.08 and 1.44, respectively. The flow rate and sample loading were then increased very significantly to evaluate the performance of the 200 mL $z^2$ cuboid SEC device under extremely challenging loading conditions. Figure 10 shows the chromatogram obtained at a flow rate of 20 mL/min by injecting 40 mL (~20% of the bed volume) of a highly concentrated feed sample, i.e., 10 g/L BSA, and 4 g/L lysozyme prepared in PBS. The resolution factor ($R$) for the BSA and lysozyme peaks was 1.35 ($\pm 0.05$). The average asymmetry factors for the BSA and lysozyme peaks were 1.11 and 1.50, respectively. Therefore, excellent resolution was obtained with the 200 mL $z^2$ cuboid SEC device, even when it was operated at extremely high flow rates with super-loaded conditions. The above separation took less than 25 min to complete, which was incredibly fast for preparative size-exclusion chromatography. A typical separation time in preparative size-exclusion chromatography is on the order of hundreds of minutes [37,38]. These results clearly demonstrate that the $z^2$ cuboid SEC device is scalable. These results also indicate that these devices represent a new way for increasing the speed, loading capacity, resolution, and scalability of preparative size-exclusion chromatography.

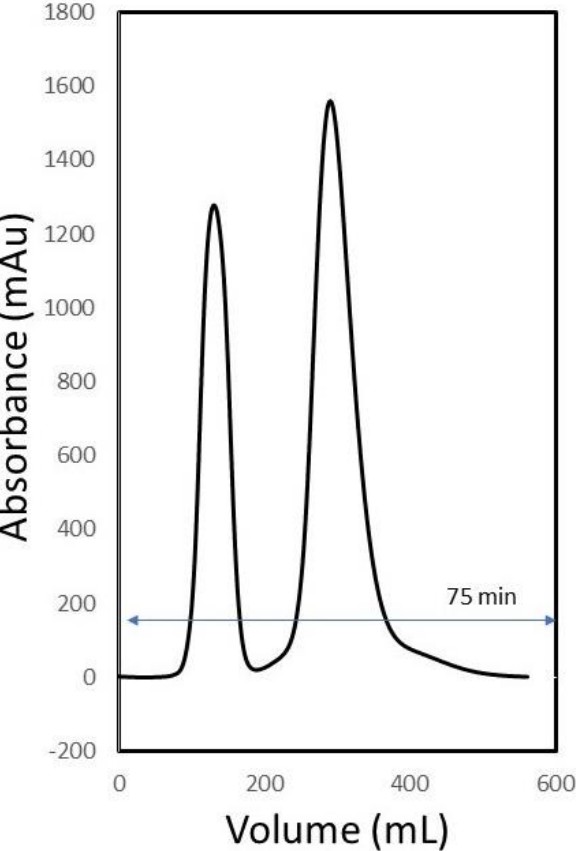

**Figure 9.** Chromatogram obtained during BSA (5 mg/mL) and lysozyme (2.5 mg/mL) separation using the 200 mL $z^2$ cuboid SEC device by loading 40 mL of feed sample at an 8 mL/min flow rate (medium: Sephacryl S-200 HR; mobile phase: phosphate-buffered saline (pH 7.4)).

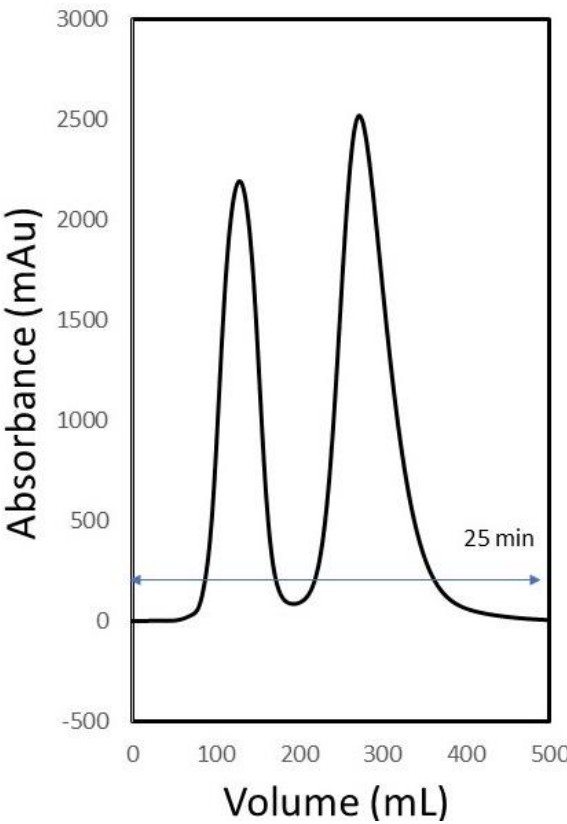

**Figure 10.** Chromatogram obtained during BSA (10 mg/mL) and lysozyme (4 mg/mL) separation using the 200 mL $z^2$ cuboid SEC device by loading 40 mL of feed sample at a 20 mL/min flow rate (medium: Sephacryl S-200 HR; mobile phase: phosphate-buffered saline (pH 7.4)).

As mentioned earlier, the use of SEC in large-scale protein purification is largely restricted to desalting and buffer-exchange-type applications due to problems with speed, capacity, and resolution. The above results demonstrate that the $z^2$ cuboid SEC device could address these challenges as it is suitable for carrying out fast, high-capacity, and high-resolution SEC of proteins. Further scale-up studies are being conducted to develop this technology further. If successful, this could lead to the wider acceptance and use of SEC in more mainstream purification of protein biopharmaceuticals. Some recent studies have shown that SEC is suitable for the purification of larger biological entities such as DNA and messenger RNA [39]; viruses used for oncolytic applications and gene therapy, bacteriophages used for treating infections [40]; exosomes and vesicles [41]; and synthetic materials such as liposomes [42], which appear in the void fraction of SEC columns. However, conventional SEC columns are severely limited due to low productivity and low scalability. The $z^2$ cuboid SEC devices discussed in this paper could potentially be used for the fast, efficient, and scalable void-fraction-targeted purification of such large species. The current study is based on one particular SEC medium, i.e., Sephacryl S-200 HR. The 200 mL $z^2$ cuboid SEC device will be tested with other SEC media, and it would be interesting to see if changing the medium has any impact on the performance for the device.

## 4. Conclusions

The $z^2$ cuboid SEC device allowed the same volume of SEC medium to be used significantly more efficiently than was possible with a conventional tall SEC column. The flow distribution and collection features of the $z^2$ cuboid SEC device ensured that the extent of the macroscale convective dispersion was minimized. Chromatographic separations could be carried out at a fraction of the pressure drop required to operate a conventional column having the same bed volume. Therefore, the $z^2$ cuboid SEC device could also

be operated at significantly higher flow rates than was possible with a conventional SEC column. At a given flow rate, the resolution in the binary protein separation obtained with the $z^2$ cuboid SEC device was significantly greater than that obtained with the conventional SEC column. Using the $z^2$ cuboid SEC device, a given target resolution could be achieved at a flow rate 2 to 3 times higher than the rate that was possible with the column. The $z^2$ cuboid SEC device was also able to handle large sample volumes, and the chromatograms obtained with this device did not show any of the typical signs of sample overloading, such as fronting and tailing. The $z^2$ cuboid SEC device is highly scalable, as indicated by the results obtained with the 200 mL device. Overall, the $z^2$ cuboid SEC device is suitable for carrying out the fast, high-capacity, high-resolution, and scalable size-exclusion chromatography of proteins and other biological macromolecules and entities.

**Author Contributions:** Conceptualization, R.G.; methodology, R.G.; software, Y.X.; validation, Y.X. and R.G.; formal analysis, Y.X. and R.G.; investigation, Y.X. and S.P.; resources, R.G.; data curation, Y.X. and R.G.; writing—original draft preparation, Y.X.; writing—review and editing, R.G.; visualization, R.G. and Y.X.; supervision, R.G.; project administration, R.G.; funding acquisition, R.G. All authors have read and agreed to the published version of the manuscript.

**Funding:** This research was funded a Discovery Grant (R.G.) from the Natural Science and Engineering Research Council (NSERC) of Canada.

**Data Availability Statement:** The data will be made available by the authors on request.

**Acknowledgments:** The authors thank the Natural Science and Engineering Research Council (NSERC) of Canada for funding this project. We thank Paul Gatt (Department of Chemical Engineering, McMaster University) for helping with fabricating the $z^2$ cuboid packed-bed devices used in this study.

**Conflicts of Interest:** The authors declare no conflict of interest.

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
