# Peer review of "A New Approach for Increasing Speed, Loading Capacity, Resolution, and Scalability of Preparative Size-Exclusion Chromatography of Proteins"

_processes, doi:10.3390/pr10122566_

Round 1
Reviewer 1 Report
In my opinion, in the presented form the manuscript (processes-2035224) entitled ‘A new approach for increasing speed, loading capacity, resolution, and scalability of preparative size exclusion chromatography of proteins’ described by Yating Xu, Si Pan and Raja Ghosh can be recommended for publication in Processes after minor revision.
My remarks and recommendations to the Processes are as follows:
The text is comprehensible.
Authors should add data regarding to:
1. Validation is missing (e.g., intra- and interday) and reproducibility.
Conclusions
‘Overall, the z2cuboid SEC device seems promising for carrying out fast, high-capacity, high-resolution and scalable size exclusion chromatography of proteins and other biological macromolecules and entities.’
The authors refer in their conclusions to proteins and other biological macromolecules.
The publication does not contain data on the impact of the matrix. It would be beneficial to supplement the data with data on the effect of the matrix. The authors should estimate the influence of the matrix and complete the data with e.g., the 'Index matrix’ or ‘matric effect'.
The z2cuboid SEC device was also able to handle large sample volumes, and the chromatograms obtained with this device did not show any of the typical signs of sample overloading such as fronting and tailing.
Please add values of symmetry of peaks (e.g., As factors).
Author Response
Reviewer 1
Comments and Suggestions for Authors
In my opinion, in the presented form the manuscript (processes-2035224) entitled ‘A new approach for increasing speed, loading capacity, resolution, and scalability of preparative size exclusion chromatography of proteins’ described by Yating Xu, Si Pan and Raja Ghosh can be recommended for publication in Processes after minor revision.
My remarks and recommendations to the Processes are as follows:
The text is comprehensible.
Response: We thanks the reviewer for the positive appraisal.
Authors should add data regarding to:
Response: We thank the reviewer for the suggestion. We have added data as suggested.
Validation is missing (e.g., intra- and interday) and reproducibility.
Response: We have added reproducibility data for the scaled up SEC device (lines 363-375).
Conclusions
‘Overall, the z2cuboid SEC device seems promising for carrying out fast, high-capacity, high-resolution and scalable size exclusion chromatography of proteins and other biological macromolecules and entities.’
Response: We thank the reviewer for the positive appraisal.
The authors refer in their conclusions to proteins and other biological macromolecules.
Response: We thank the reviewer for the suggestion. On line 399-401, we have listed other biological macromolecules that could potentially be purified using such devices.
The publication does not contain data on the impact of the matrix. It would be beneficial to supplement the data with data on the effect of the matrix. The authors should estimate the influence of the matrix and complete the data with e.g., the 'Index matrix’ or ‘matric effect'.
Response: We thank the reviewer for the suggestion. Our experiments were carried out using one particular type of matrix, i.e., Sephacryl S 200 HR. So, we were not able to compare different matrices in our current study. However, we intend to test our device with other matrices and have added this plan in the revised manuscript. We have clarified this in line 405-408.
The z2cuboid SEC device was also able to handle large sample volumes, and the chromatograms obtained with this device did not show any of the typical signs of sample overloading such as fronting and tailing.
Response: We thank the reviewer for highlighting the positive attribute of your device.
Please add values of symmetry of peaks (e.g., As factors).
Response: We thanks the reviewer for the suggestion. We have added asymmetry factor data in the revised manuscript (lines 368-376).
Reviewer 2 Report
The manuscript presents a novel approach to carrying out size exclusion chromatography. In my opinion, the work is original, and the presented results are really interesting.
If we assume that it is only the presentation of the idea, I have only a minor comment on this work:
What about moving Fig. 3 and Fig. 4 to supplement?
However, if we look at the topic deeper some questions arise:
Some typical calibration curve and elution data may be interesting (a series of peak profiles).
Some explanation, or some idea of explanation, why the z2 is better than short and wide column may be nice addition to the conclusion.
Author Response
Reviewer 2
Comments and Suggestions for Authors
The manuscript presents a novel approach to carrying out size exclusion chromatography. In my opinion, the work is original, and the presented results are really interesting.
Response: We thank the reviewer for the positive and supportive comment.
If we assume that it is only the presentation of the idea, I have only a minor comment on this work:
What about moving Fig. 3 and Fig. 4 to supplement?
Response: We thank the reviewer for the suggestion. We agree that two separate figures showing the devices is superfluous. Therefore, we have combined Figures 3 and 4 in one, i.e., new Figure 3. The subsequent figures are renumbered accordingly.
However, if we look at the topic deeper some questions arise:
Some typical calibration curve and elution data may be interesting (a series of peak profiles).
Response: We thank the reviewer for the suggestion. We have added peak profiles showing a systematic calibration study on the effect of flow rate on performance (see new Figure 6).
Some explanation, or some idea of explanation, why the z2 is better than short and wide column may be nice addition to the conclusion.
Response: We thank the reviewer for the suggestion. We have added a detailed explanation on why the z2 is better than short and wide column (line 239-250).